# OpenReview forum: "Multiscale‑Coherent Representations of Alzheimer’s Disease Reversal via Agentic Active Causal Discovery"
_ICLR.cc/2026/Workshop/LMRL — Submitted to ICLR 2026 Workshop LMRL_

### Official Review · Reviewer_ZPAH · 2026-02-23

**Rating:** 7
**Confidence:** 4

**Review:**

This paper proposes a framework for learning causal, multiscale-coherent representations for Alzheimer’s disease reversal, aiming to capture interventional structure across biological scales rather than purely correlational patterns. The approach combines multiscale representation learning with an active causal discovery loop and a proxy validation protocol based on wavelet coherence.

The paper addresses an interesting and relevant problem for the workshop, but my main concern is that it feels over-scoped relative to the concrete evidence provided, and it is difficult to follow on a first read e.g. several figures intended to clarify the pipeline actually add confusion. The method involves many components that are described at a high level, but it is not always clear how each part is implemented and how the full system is evaluated end-to-end. For example, while the wavelet-coherence proxy is an interesting diagnostic, its connection to causal correctness or biological meaningfulness is not convincingly justified. Moreover, the reported NOTEARS results show very low recall and F1, which weakens the empirical support for the claims. Overall, the paper would benefit from tighter experimental reporting (datasets, preprocessing, baselines, seed variability) and a clearer separation between the conceptual proposal and what is actually done and validated in this submission.

Despite the above issues, the paper is well aligned with the workshop theme (meaningful representations of life) through its focus on causal, multiscale representations and disease reversal as a motivating application. I found it an interesting submission for the workshop even if the current version needs clearer presentation and stronger, more targeted validation. For these reasons, I lean toward the acceptance of the paper.

---

### Meta-Review · Area_Chair_VVSD · 2026-02-28

**Recommendation:** Reject
**Confidence:** 4

**Metareview:**

Although the review for this paper is generally positive, some clear concerns were outlined; the paper is hard to follow and lacks sufficient information for experimental details. Moreover, section 5.3 is titled "VALIDATION EXPERIMENTS REVIEWERS WILL EXPECT". As such, this paper is recommended for rejection.

---

### Decision · Program_Chairs · 2026-03-02

**Decision:**

Reject

**Comment:**

Please see the meta-review.